# Impairment on Cardiovascular Autonomic Modulation in Women with Migraine

**DOI:** 10.3390/ijerph20010763

**Published:** 2022-12-31

**Authors:** Denise Martineli Rossi, Hugo Celso Dutra de Souza, Débora Bevilaqua-Grossi, Ana Carolina Carmona Vendramim, Stella Vieira Philbois, Gabriela Ferreira Carvalho, Fabíola Dach, Sérgio Mascarenhas, Anamaria Siriani de Oliveira

**Affiliations:** 1Department of Applied Physiotherapy, Federal University of Triângulo Mineiro, Uberaba 38025-180, MG, Brazil; 2Department of Health Sciences, Ribeirão Preto Medical School, University of São Paulo, Ribeirao Preto 14049-900, SP, Brazil; 3Department of Physiotherapy, Institute of Health Sciences, Universität zu Lübeck, 23562 Lübeck, Germany; 4Department of Neurosciences and Behavioral Sciences, Ribeirao Preto Medical School, University of São Paulo, Ribeirao Preto 14049-900, SP, Brazil; 5São Carlos Institute of Physics, University of São Paulo, Sao Carlos 13566-590, SP, Brazil

**Keywords:** headache, heart rate variability, blood pressure variability

## Abstract

Autonomic dysfunction, such as reduced vagally mediated heart rate variability, has been suggested in headache patients but is still uncertain when considering primary headache disorders. This study aims to compare the heart rate and blood pressure variability and baroreflex sensitivity between women with migraine and controls. A migraine (n = 20) and a control group (n = 20) of age-matched women without headache were evaluated. Heart rate variability was analyzed through frequency-domain using spectral analysis presenting variance, low-frequency (LF; 0.04–0.15 Hz) and high-frequency (HF; 0.15–0.4 Hz) bands and by time domain (root mean square of successive R-R interval differences, RMSSD). Blood pressure variability was analyzed with spectral analysis and baroreflex sensitivity with the sequence method. Migraine group had lower heart rate variability characterized by a reduction in total variance, LF oscillations (sympathetic/vagal modulation) and HF oscillations (vagal modulation), and a reduction in SD and RMSSD compared to control group. No difference was found in the blood pressure variability analysis. Regarding baroreflex sensitivity, migraine group had decreased values of total gain, gain down and up compared to control group. Women with migraine exhibited autonomic modulation alterations, expressed by decreased values of heart rate variability and baroreflex sensitivity, but not by differences in blood pressure variability.

## 1. Introduction

Autonomic nervous system dysfunction has been linked to common headache disorders, including migraine, which plays an important role in the understanding of many characteristic symptoms and signs of this disease with complex pathophysiology [1,2,3]. The decrease in vagal influence is a risk factor for all-cause mortality and for cardiovascular diseases and is related to greater inflammation and less inhibitory control of pain [4,5]. The literature [6] suggests a lower vagal modulation in headache patients, including migraine, cluster, and tension-type headaches, by means of a time domain analysis of the root mean square of successive R-R interval differences (RMSSD) [6].

In migraine patients, a systematic review with meta-analysis [7] suggested a decrease in sympathetic and parasympathetic modulation, observed by increased P-wave dispersion, longer QTc intervals lengths, and a decrease in the rate of deep breathing in patients with migraine, but without significant differences in the values of root mean square of successive R-R interval differences (RMSSD), standard deviation normal to normal (SDNN), R-R interval and Valsalva ratio [7]. Episodic migraine patients, in the ictal period, present a decrease in parasympathetic modulation observed by a decrease in standard deviation normal to normal, i.e., time domain analysis, and in the sympathetic modulation observed by low-frequency (LF), i.e., frequency domain analysis [8].

Additionally, baroreflex sensitivity is considered a robust and interpretable measure of balance (health) and the effectiveness of cardiovascular autonomic control [9]. Impairments in the sympathetic/vagal autonomic drive, present in different pathophysiological situations, are accompanied by a reduction in baroreflex sensitivity [10]. Nilsen et al. (2009) compared young women with migraine to asymptomatic controls and found increases in baroreflex sensitivity and heart rate variability, suggesting that central hypersensitivity in migraine also includes cardiovascular reactivity [9]. Furthermore, an increased blood pressure variability has also been suggested in episodic migraine patients, which shows a vascular reactivity mediated by sympathetic efferent nerves [2].

The current knowledge is still unable to determine the altered autonomic modulation when considering different diagnoses of primary headache disorders based on well-established criteria, such as migraine, and by considering homogenous samples by age and sex [5]. Thus, this study aimed to compare the heart rate variability, blood pressure variability, and baroreflex sensitivity between women with migraine and age-matched women controls. We hypothesized an impairment in cardiovascular autonomic control in migraine patients compared to controls.

## 2. Materials and Methods

### 2.1. Participants

The study sample involved 42 women, including a migraine group (MG, n = 22) and a control group (CG, n = 20), age-matched at 29.6 (9.4) years. All migraine patients involved in this study were recruited from a tertiary headache clinic and were included following the International Classification of Headache Disorders (ICHD-III) criteria for migraine by an experienced neurologist. Patients with other secondary headaches were excluded. Women included in the CG had no history of current or past primary headache diagnosis and no history of recurrent secondary headaches in the last 6 months that required treatment. All participants, patients, and health volunteers were between 18 and 50 years old and were excluded in the case of pregnancy, diabetes mellitus, arterial hypertension, cardiovascular diseases, habitual use of corticosteroids, and history of alcohol abuse in the last 6 months. We also excluded patients in menopause [11] and those who were taking the medicine venfalaxine [12] due to their known effects on heart rate variability.

Anthropometric characteristics and basic clinical data were gathered from both groups using validated scales such as the Pain Catastrophizing Scale [13], Central Sensitization Inventory [14], and Beck Depression Inventory [15] were also evaluated. Moreover, clinical characteristics of the migraine were also evaluated as the frequency of migraines, years with the symptoms and attach duration (years), pain intensity, and migraine disability evaluated by the Migraine Disability Assessment [16], and allodynia assessed by the 12-item allodynia symptom checklist [17].

The Institutional Ethics Committee approved the study (protocol number: 1278/2019), and written informed consent was obtained from all participants prior to data collection.

### 2.2. Data Collection

Autonomic function tests were carried out in the Exercise Physiology and Cardiovascular Physiotherapy Laboratory, Department of Health Sciences, Ribeirão Preto Medicine School University, under standardized conditions. All women were evaluated in the follicular phase of the regular menstrual cycle and during the afternoon. They were instructed to avoid intense exercise a day before the test and to avoid drinking alcohol and caffeine. Volunteers were instructed to remain in the supine position for approximately 10 min to stabilize cardiovascular parameters. After this period, the ECG and arterial pulse pressure were recorded simultaneously for another 10 min.

The heart rate data were obtained with an ECG digital recorder (ML866 PowerLab, ADInstruments, Bella Vista, Australia) through the modified MC5 shunt at a sampling frequency of 1000 Hz. The systolic blood pressure was obtained using digital plethysmography recording equipment (Finometer Pro, Finapres Medical System, Amsterdam, Netherlands) using a cuff positioned on the finger of the right upper limb. The data were recorded with an interface to the microcomputer using the PowerLab, LabChart version 8 (ADinstruments Pty Ltd., Bella Vista, NSW, Australia).

### 2.3. Data Analysis

The heart rate variability and the blood pressure variability analyses were performed with custom computer software (CardioSeries version 2.7, http://sites.google.com/site/cardioseries (accessed on 20 January 2022)) developed by Dias, DPM of the University of São Paulo, Brazil [18]. The heart rate variability was analyzed with two methods: time domain (standard deviation, SD, and RMSSD) and frequency domain (spectral analysis, Fast Fourier Transform) (Task Force of the European Society of Cardiology, the North American Society 1996). Frequency domain analysis breaks down heart rate variability into fundamental oscillatory components, the main ones being the high-frequency (HF), which is associated with increased parasympathetic tone, and the low-frequency (LF), which is associated with increased sympathetic tone [3]. The blood pressure variability was analyzed in the frequency domain, and the baroreflex sensibility was assessed in the time domain using the sequence technique [19].

For spectral analysis, the values of R-R intervals (RRi) were redesigned in a 3 Hz cubic spline interpolation to normalize the time interval between the beats. The series of interpolated RRi and SBP were divided into half-overlapping sets of 256 data points, overlapping 50% (Welch Protocol). The stationary segment was visually inspected, and those with artifacts or transients were excluded. Each RRi and SBP stationary segment was subjected to spectral analysis using Fast Fourier Transform after the Hanning window. RRi time series were integrated with bands of low-frequency (LF; 0.04–0.15 Hz) and high-frequency (HF; 0.15–0.4 Hz), and the results were expressed in absolute (ms^2^) and normalized units (nu), whereas the SBP time series were integrated only in the low-frequency band (LF; 0.04–0.15 Hz), and the results were expressed in absolute values (mmHg^2^).

The normalized values for heart rate variability were obtained by calculating the percentage of LF and HF power in relation to the total spectrum power minus the very low-frequency band (VLF; <0.04 Hz). In addition, the normalization procedure was performed to minimize the total power variations in the absolute values of LF and HF (Task Force of the European Society of Cardiology, the North American Society 1996). To assess the sympathovagal balance, the LF/HF ratio of RRi variability was also calculated [20].

Baroreflex sensitivity was assessed in the time domain using the sequence technique [19]. The computer software CardioSeries version 2.7 scanned the beat-to-beat time series of pulse interval (PI) and SBP values, searching for sequences of at least 3 consecutive beats in which progressive increases in SBP were followed by progressive increases in PI (up sequence) and progressive decreases in SBP were followed by progressive decreases in PI (down sequence), with a correlation coefficient (r) between PI and SBP (values higher than 0.8). Spontaneous baroreflex sensitivity was determined by the mean slope of the linear regression between the SBP and PI values of each sequence. The sequence method also presents the baroreflex effectiveness index (BEI). It is the ratio of the number of sequences and the total number of SBP ramps. The BEI shows how many SBP changes are effectively translated into a change in PI, independent of its magnitude [21].

### 2.4. Statistical Analysis

The sample size was based on a pilot study with 10 participants in each group considering the RMSSD difference between the migraine group (mean: 30.86, SD: 17.44) and control group (mean: 51.62, SD: 29.42) and an effect size of 0.88.

The power and the alpha level were set at 85% and 5%, respectively, resulting in a minimum of 20 participants in each group.

Age, body mass index, and total score of the pain catastrophizing, central sensitization, and depression questionnaires were compared between groups by using the independent sample *t*-test. All between-group comparisons were performed using the independent Student’s t-test after log transformation when normal distribution was not verified (Shapiro–Wilk *p* > 0.05). The effect sizes (Cohen’s d) were calculated by dividing the difference between the means of the two groups by their pooled standard deviation [22] and interpreted as follows: d ≤ 0.5: small; 0.5 < d ≤ 0.8: moderate; and d > 0.8: large. All statistical analyses were completed using SPSS software, version 20 (SPSS Inc., Chicago, IL, USA), and a significance level of 0.05 was established.

## 3. Results

Women with migraine had a headache frequency of 14.2 (6.8) days per month being of high frequency (8 to 15 attacks per month, n = 11) [23] and chronic migraine patients (more than 15 attacks per month, n = 09), migraine onset of 9.2 (7.0) years and showed severe pain intensity during an attack (Table 1). Other clinical characteristics of migraine are also shown in Table 1. Women with migraine showed increased scores in pain catastrophizing (*p* < 0.01, t = 3.963), central sensitization (*p* < 0.01, t = 4.336), and depression (*p* < 0.01, t = 3.208) questionnaires (Table 1). Regarding the type of medication, 45.0% of women in the migraine group self-reported the use of analgesics, 45.0% of antidepressants, 25.0% of anti-inflammatory, 30.0% of antipsychotics, 5% of anxiolytics, and 30.0% of antiepileptic drugs.

Compared to controls, women with migraine had reduced heart rate variability and baroreflex sensitivity with a moderate effect size (Table 2).

Analysis of heart rate variability in the frequency domain showed lower variance (*p* = 0.007, t = −2.85), reduced vagal autonomic influence evidenced by the lower HF band (*p* = 0.008, t = −2.78), and a decrease in sympathetic autonomic influence evidenced by the lower LF band (*p* = 0.014, t = −2.589) in the migraine group compared to the control group. Frequency domain analysis using spectral analysis (linear analysis) decomposes heart rate variability into fundamental oscillatory components, the main ones being the high-frequency (HF) band, which is associated with vagal autonomic modulation, and the low-frequency (LF) band, which is associated with both sympathetic and vagal modulation [3,24].

Additionally, heart rate variability analysis in the time domain showed decreased vagal modulation expressed by SD (*p* = 0.008, t = −2.81) and RMSSD (*p* = 0.004, t = −3.04) with a large effect size in women with migraine compared to controls. The variables analyzed in the time domain, such as RMSSD, when reduced, are correlated with generalized autonomic impairment and increased morbidity and mortality [3,24].

On the other hand, the LF and HF oscillations in normalized units and the LF/HF ratio did not show significant differences between groups (Table 2). The LF/HF ratio represents the sympathovagal balance under the heart, with a higher LF/HF suggesting greater sympathetic impulse and a lower one suggesting greater parasympathetic impulse [3,24].

In blood pressure variability, variance and absolute values of LF oscillations were not different between groups. Regarding baroreflex sensitivity, the migraine group had decreased values of total gain (*p* = 0.008, t = −2.77), gain down (*p* = 0.003, t = −3.20), and gain up (*p* = 0.002, t = −3.27) compared to controls.

## 4. Discussion

This cross-sectional study found a decreased heart rate variability and reduced vagal and sympathetic autonomic influence in women with high-frequency and chronic migraine compared to age-matched controls. The findings were suggested and expressed in the analysis of heart rate variability in the frequency domain, by the fluctuation of the LF and HF values, and in the time domain, by the SD and RMSSD values with large effect sizes. The migraine group also showed decreased values of baroreflex sensitivity, and there were no between-group differences in the blood pressure variability.

In the context of existing knowledge, migraine appears to be associated with autonomic dysfunction, although the source of this is still uncertain. Impaired vagal function, as found in patients with migraine in this study, has been associated with exacerbation of inflammatory processes. Parasympathetic tone acts on the cholinergic anti-inflammatory pathway by releasing acetylcholine into the reticuloendothelial system, which inhibits the release of pro-inflammatory cytokines but does not inhibit anti-inflammatory cytokines, reflecting a malfunctioning anti-inflammatory reflex (23). These inflammatory processes can cause continuous and prolonged excitation of primary nociceptive neurons, leading to chronic painful conditions (5). Additionally, decreased parasympathetic tone results in greater somatic and visceral input through the spinothalamic pathway, lowering the pain threshold in those with chronic pain (6). However, the participation of calcitonin gene-related peptide (cGRP) in this process cannot be ruled out. cGRP is a 37 amino acid neuropeptide known to be a vasodilator [25] expressed in both the central and peripheral nervous systems [26,27]. cGRP is associated with autonomic cardiovascular regulation through its action at different sites of cardiovascular regulation, such as the sinoatrial and atrioventricular nodes, the nucleus of the solitary tract, and the hypothalamus, which results in an important influence on the sympathetic and vagal autonomic tonus [28]. However, the negotiation by which CGRP would play its role in cardiovascular regulation remains controversial.

A more pronounced heart rate variability reduction was suggested in episodic migraine patients during the ictal period with a reduced LF [8]. On the other hand, a recent meta-analysis [7] analyzed the RMSSD and standard deviation of the R-R intervals (SDNN) values and found no differences between migraine patients and controls. However, the heart rate variability parameters in the frequency domain, blood pressure variability, and baroreflex sensitivity were not analyzed, which is one of the differentials of our study. Lee et al. (2019), nevertheless, suggest a change in nerve conductance to the heart due to an imbalance in the autonomic nervous system, which would promote a longer QTc interval and increase the risk of arrhythmias in patients with migraine [7]. In contrast, the meta-analysis by Koenig et al. (2015), when analyzing the effect of the RMSSD values, indicated less vagal modulation in patients with headache; however, they reinforced the importance of further studies with more homogeneous samples [5]. In the sub-analyses of the baroreflex sensitivity, our study obtained different results from previous studies. Nilsen et al. (2009) observed an increase in baroreflex sensitivity in migraine patients compared to controls, while in our study, this variable was significantly lower in the migraine group [9].

The present study has some limitations. Our study did not use autonomic cardiovascular reflex tests to assess the autonomic nervous system; however, the analysis of the variability of heart rate, blood pressure, and baroreflex sensitivity has been considered an effective, sensitive, and non-invasive method of measuring autonomic impulses. In addition, the sample size of our study may not have power for all variables, as it was previously calculated based on RMSSD. Moreover, our results should be restricted to patients with migraine headache more than 8 days per month with or without aura since we could not conduct separate sub-analyzes considering the effect of the presence of aura and possible differences between patients with chronic and episodic migraine.

The women with migraine in our study had increased pain catastrophizing, central sensitization, and depression which is in accordance with a recent meta-analysis from Caponnetto et al. (2021) showing the high prevalence of depressive disorders, fibromyalgia, and musculoskeletal pain, mainly in women [29]. Although the current study investigated only women with migraine, which is three times more common than in men, a better understanding of cardiovascular autonomic modulation in men is also needed in the literature, considering that sex hormones may act as important modulators [30]. In this context, our study analysis could not consider the medication effect on the investigated variables, but we have presented the self-reported medications in use by the patients, and we excluded the patients using venfalaxine.

The decrease in vagal influence is a risk factor for all-cause mortality and for cardiovascular diseases and is related to greater inflammation and less inhibitory control of pain [4,5]. In this sense, our study contributes to a better understanding of the cardiac autonomic modulation in women with high-frequency and chronic migraine compared to age-matched controls by controlling for age, menstrual period, period of the day of the evaluation, caffeine, and alcohol intake [5,31].

## 5. Conclusions

Women with migraine exhibit autonomic modulation alterations, expressed by decreased values of heart rate variability and baroreflex sensitivity, but not by differences in blood pressure variability. Our results should be restricted to patients with migraine headache more than 8 days per month with or without aura since we could not conduct a separate sub-analysis.

## Figures and Tables

**Table 1 ijerph-20-00763-t001:** Sample characteristics.

	Migraine Group (n = 20)	Control Group (n = 20)
Age (years)	29.6 (10.3)	29.6 (9.4)
BMI (kg/m^2^)	25.3 (3.6)	22.9 (2.8)
Migraine frequency (monthly)	14.2 (6.8)	N/A
Chronic migraine (%)	9 (45%)	N/A
Migraine onset (years)	9.2 (7.0)	N/A
Attack duration (hours)	12.4 (14.5)	N/A
Aura	50% (10)	N/A
Pain intensity (NRS: 0 to 10)	7.8 (1.6)	N/A
Disability (MIDAS)	43.4 (29.8)	N/A
Allodynia (ASC-12)	5.3 (2.3)	N/A
Pain catastrophizing (PCS)	23.5 (11.8)	8.5 (10.8) *
Central sensitization (CSI-BR)	40.4 (15.7)	22.3 (9.2) *
Depression (BDI)	14.6 (11.6)	4.8 (5.6) *

BMI, body mass index; NRS, numeric rating scale; MIDAS, Migraine Disability Assessment; ASC-12, 12-item allodynia symptom checklist; PCS, Pain Catastrophizing Scale; CSI-BR (Central Sensitization Inventory-BR); BDI (Beck Depression Inventory-II). Values are presented as mean (SD) and percentage (n). * *p* < 0.05 for between-group differences.

**Table 2 ijerph-20-00763-t002:** Heart rate variability, blood pressure variability, and spontaneous baroreflex sensitivity in migraine and control groups.

	Migraine Group (n = 20)	Control Group (n = 20)	Mean Difference (IC 95%)	Effect Size
Heart rate variability				
RRi	815.9 (120.6)	881.3 (91.7)	−65.4 (−133.9; 3.2)	−0.61
Frequency domain				
variance	1653.9 (1459.6)	3163.5 (2776.4)	−1509.6 (−2929.5; −89.7) *	−0.68
LF abs, (ms^2^)	499.1 (499.5)	1098.9 (1336.5)	−599.5 (−1244.9; 45.9) *	−0.59
HF abs, (ms^2^)	498.1 (458.8)	1240.7(1423.6)	−742.5 (−1436.6; −48.5) *	−0.70
LF/HF	1.30 (0.8)	1.2 (0.7)	0.1 (−0.4; 0.6)	0.13
LF nu	46.9 (12.1)	47.2 (14.9)	0.9 (−7.7; 9.5)	−0.02
HF nu	53.1 (12.1)	52.8 (14.9)	−0.9 (−9.5; 7.7)	0.02
Time domain				
SD	41.3 (17.8)	57.5 (22.3)	−16.2 (−29.2; −3.3) *	−0.80
RMSSD	30.5 (15.9)	49.5 (27.1)	−18.9 (−33.2; −4.7) *	−0.86
Blood pressure variability				
Variance (mmHg^2^)	19.3 (16.0)	14.9 (9.4)	4.4 (−4.0; 12.8)	0.34
LF abs (mmHg^2^)	5.1 (5.5)	4.8 (2.9)	0.4 (−2.4; 3.2)	0.07
Baroreflex sensitivity				
BEI	0.6 (0.3)	0.7 (0.3)	−0.1 (−0.3; 0.1)	−0.33
Baroreflex sequences (n)	105.6 (50.8)	106.8 (54.9)	−1.2 (−35.1; 32.7)	−0.02
All Gain (ms/mmHg)	10.3 (6.1)	16.5 (8.9)	−6.2 (−11.1; −1.3) *	−0.81
Gain down (ms/mmHg)	10.4 (6.0)	16.0 (7.4)	−5.6 (−9.9; −1.3) *	−0.83
Gain up (ms/mmHg)	10.1 (6.4)	16.9 (10.2)	−6.8 (−12.2; −1.3) *	−0.80

RRi, interval between the R-R waves; HF, high frequency; LF, low frequency; BEI, baroreflex effectiveness index; ms^2^, milliseconds; nu, normalized units; SD, standard deviation; RMSSD, root mean square of successive differences; ms/mmHg, milliseconds/millimeter of mercury; up, baroreflex sequence with progressive increases in blood pressure followed by progressive increases in pulse interval; down, baroreflex sequence with progressive decreases in blood pressure followed by progressive decreases in pulse interval. * *p* < 0.05 for between-group differences.

## Data Availability

The authors confirm that the data supporting the findings of this study are available within the article.

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
