# Peer review of "Impairment on Cardiovascular Autonomic Modulation in Women with Migraine"

_ijerph, 2022, doi:10.3390/ijerph20010763_

Round 1

Reviewer 1 Report

Very well designed research article on the cardiovascular autonomic regulation in women affect by migraine, studying parameters associated to autonomic modulation like heart rate variability and blood pressure variability.

Overall the article is well written, the methods are well described, the sections and subsections are clear.

I have only minor comments.

Line 33: add more relevant keywords (e.g. heart rate variability, …).

Line 46: remove lengths.

Line 54: the authors say: “the baroreflex sensitivity is considered a robust and interpretable measure”. Of what?

Line 68: it sound better “age-matched”.

Line 83: venlafaxine is not an hormone but an antidepressant (SNRI).

Line 85-86: please provide here the name of the scale used, and the appropriate reference.

Line 159: please correct “larg.”.

Table 1. Provide the meaning of PCS and CSI-BR in the footnote.

Table 2: adjust the formatting of the footnote, and correct the spacing with the main text.

Line 190-191: reference style has changed. Provide the appropriate numbers.

Line 197: reference style has changed. Provide the appropriate numbers.

Line 202: reference style has changed. Provide the appropriate numbers.

Line 216-226: since the recent approval of anti-cGRP antibodies, the authors should add a brief explanation of how the autonomic dysfunction is related with the cGRP.

247: change to “sub-analyses”.

Line 285-294: remove the section “appendix A and B” since there are no data.

Author Response

Response to Reviewer 1 Comments

The authors would like to thank the editorial team for the opportunity to resubmit this work. We would also like to thank the reviewers for their reviews, which certainly helped to improve the quality of the manuscript. We carefully considered all the comments, suggestions, and corrections, and we have made appropriate changes to the manuscript.

Reviewer 1

Comments and Suggestions for Authors

Very well designed research article on the cardiovascular autonomic regulation in women affect by migraine, studying parameters associated to autonomic modulation like heart rate variability and blood pressure variability. Overall the article is well written, the methods are well described, the sections and subsections are clear.

I have only minor comments.

Point 1. Line 33: add more relevant keywords (e.g. heart rate variability, …).

Response 1: The authors have changed all keywords based on the reviewer suggestion: Keywords: headache, heart rate variability, blood pressure variability.

Point 2. Line 46: remove lengths.

Response 2: The authors have changed the sentence according to the reviewer suggestion:

Lines 45: “In migraine patients, a systematic review with meta-analysis (1) suggested decrease in sympathetic and parasympathetic modulation..”

Point 3. Line 54: the authors say: “the baroreflex sensitivity is considered a robust and interpretable measure”. Of what?

            Response 3: The authors have clarified this sentence:

Line 54: “Additionally, the baroreflex sensitivity is considered a robust an interpretable measure of balance (health) and effectiveness of cardiovascular autonomic control (9,10). Impairments in the sympathetic/vagal autonomic drive, present in different pathophysiological situation, are accompanied by a reduction in baroreflex sensitivity (3).

Point 4. Line 68: it sound better “age-matched”.

Response 4: The authors have changed the sentence as suggested.

Line 67: “..with migraine and age-matched women controls...”

Point 5. Line 83: venlafaxine is not an hormone but an antidepressant (SNRI).

Response 5: We appreciate the reviewer for this correction. We have corrected it.

Point 6. Line 85-86: please provide here the name of the scale used, and the appropriate reference.

Response: 6 We have added all scales and questionnaires’ names and their validity studies references.

Line 85: “Anthropometric characteristics and basic clinical data from both groups by using validated scales as: Pain Catastrophizing Scale (4), Central Sensitization Inventory (5), Beck Depression Inventory (6) were also evaluated. Moreover, clinical characteristics of the migraine were also evaluated as frequency of migraines, years with the symptoms and attach duration (yours), pain intensity, and migraine disability evaluated by the Migraine Disability Assessment (7) and allodynia by the 12-item allodynia symptom checklist (8)”.

Point 7. Table 1. Provide the meaning of PCS and CSI-BR in the footnote.

Response 7: We have added the information on Table 1 as suggested.

Point 8. Table 2: adjust the formatting of the footnote, and correct the spacing with the main text.

Response 8: We have corrected the Table 2 as suggested.

Point 9. Line 190-191: reference style has changed. Provide the appropriate numbers. Line 197: reference style has changed. Provide the appropriate numbers. Line 202: reference style has changed. Provide the appropriate numbers.

Response 9: Done.

Point 10. Line 216-226: since the recent approval of anti-cGRP antibodies, the authors should add a brief explanation of how the autonomic dysfunction is related with the cGRP.

Response 10: We appreciate the suggestion. We have inserted a brief explanation about the autonomic dysfunction and the relationship with cGRP (Page 7, 3rd paragraph, line 226).

Line 234: “However, the participation of calcitonin gene-related peptide (cGRP) in this process cannot be ruled out. cGRP is a 37 amino acid neuropeptide known to be a vasodilator (Brain & Williams, 1985) expressed in both the central and peripheral nervous systems (Zaidi et al., 1985; Hokfelt et al., 1992). cGRP is associated with autonomic cardiovascular regulation through its action at different sites of cardiovascular regulation, such as the sinoatrial and atrioventricular nodes, the nucleus of the solitary tract and the hypothalamus, which would result in an important influence on the sympathetic and vagal autonomic tonus (Bell & McDermott, 1996; Mai et al., 2014). However, the negotiation by which CGRP would play its role in cardiovascular regulation remains controversial”.

Zaidi M, et al. Circulating CGRP comes from the perivascular nerves. Eur J Pharmacol. 1985;117(2):283–4. [PubMed: 3878299]

Hokfelt T, et al. Calcitonin gene-related peptide in the brain, spinal cord, and some peripheral systems. Ann N Y Acad Sci. 1992; 657:119–34. [PubMed: 1637079]

Bell D, McDermott BJ. Calcitonin gene-related peptide in the cardiovascular system:

characterization of receptor populations and their (patho)physiological significance. Pharmacol Rev. 1996; 48(2):253–88. [PubMed: 8804106]

Mai TH, Wu J, Diedrich A, Garland EM, Robertson D. Calcitonin gene-related peptide (CGRP) in autonomic cardiovascular regulation and vascular structure. J Am Soc Hypertens. 2014 May;8(5):286-96. doi: 10.1016/j.jash.2014.03.001. Epub 2014 Mar 13.

Point 11. 247: change to “sub-analyses”.

Response 11: We have changed the term as suggested.

Point 12. Line 285-294: remove the section “appendix A and B” since there are no data.

Response 12: We have removed as suggested.

Reviewer 2 Report

This original study aims to compare heart rate variability, blood pressure variability and baroreflex sensitivity between women with migraine and age-matched control group. The authors hypothesized an impairment on  cardiovascular control in migraine patients.

The manuscript is written with a clear, scientific language and easy to follow. The introduction is clear and methods are explained well. The study design is relevant and results have been presented clearly. The discussion is reasonable including the limitations.

The major issue is that the sample size is too small, therefore the authors should be humble to draw conclusions. Another issue is the heterogeneity of the study group. To this reviewer it is not clear what percentage of migraineurs have chronic and episodic migraine. This difference might have great impact on the results. Moreover there are others factors that need to be discussed such as effect of medications, migraine with aura vs. without aura and migraine in men. Without doing subgroup analysis it will be very difficult the generalize the results. 

Author Response

Response to Reviewer 1 Comments

The authors would like to thank the editorial team for the opportunity to resubmit this work. We would also like to thank the reviewers for their reviews, which certainly helped to improve the quality of the manuscript. We carefully considered all the comments, suggestions, and corrections, and we have made appropriate changes to the manuscript.

Reviewer 2

Comments and Suggestions for Authors

This original study aims to compare heart rate variability, blood pressure variability and baroreflex sensitivity between women with migraine and age-matched control group. The authors hypothesized an impairment on cardiovascular control in migraine patients. The manuscript is written with a clear, scientific language and easy to follow. The introduction is clear and methods are explained well. The study design is relevant and results have been presented clearly. The discussion is reasonable including the limitations.

Point 1. The major issue is that the sample size is too small, therefore the authors should be humble to draw conclusions. Another issue is the heterogeneity of the study group. To this reviewer it is not clear what percentage of migraineurs have chronic and episodic migraine. This difference might have great impact on the results.

Response 1: The authors appreciate this relevant suggestion. The sample size of 20 participants in each group was based on a power calculation with beta: 85% and alpha: 5% to estimate group differences for the RMSSD variable with an effect size of 0.88. After your valuable suggestion, we opted to exclude two participants which had a migraine frequency of 4 and 5 episodes per month. In this way, our sample (20 participants per group) was more homogeneous by being composed of high-frequency (8 to 14 attacks per month, n=11) and chronic migraine patients (more than 15 attacks per month, n=09) (Barbanti et al., 2022; Jedynak et al., 2021). We have added this information in the limitation section:

Line 262: “In addition, the sample size of our study may not have power for all variables, as it was previously calculated based on RMSSD. Moreover, our results should be restricted to patients with migraine headache higher than 8 day per month with or without aura once we could not do separate sub-analyzes considering the effect of presence of aura and possible differences between patients with chronic and episodic migraine”.

We also rewrote the conclusion based on your suggestion:

Line 287: “Women with migraine exhibit autonomic modulation alterations, expressed by decreased values of heart rate variability and baroreflex sensitivity, but not by differences on blood pressure variability. Our results should be restricted to patients with migraine headache higher than 8 day per month with or without aura once we could not do separate sub analysis”.

Point 2. Moreover there are others factors that need to be discussed such as effect of medications, migraine with aura vs. without aura and migraine in men. Without doing subgroup analysis it will be very difficult the generalize the results. 

Response 2: According to the medication, we have included the medication intake by the participants in the results section and also in the discussion section:

Line 172: “Regarding the type of medication, 45.0% of women in migraine group self-reported the use of analgesics, 45.0% of antidepressants, 25.0% of anti-inflammatory, 30.0% of antipsychotics, 5% of anxiolytics and 30.0% of antiepileptic drugs”.

Line 268: “The women with migraine in our study had increased pain catastrophizing, central sensitization and depression which is in accordance with a recent meta-analysis from Caponnetto et al. (2021) showing the high prevalence of depressive disorders, fibromyalgia and muscuskeletal pain mainly in women (Caponneto et al., 2021). Although the current study has investigated only women with migraine which is three more common than in men, a better understanding of cardiovascular autonomic modulation in men is also needed in the literature considering that sex hormones may act as important modulators (Martelletti, 2022). In this context our study analysis could not consider the medication effect on the investigated variables, but still we have presented the self-reported medications in use by the patients, and we excluded the patients in use of venfalaxine”.

Caponnetto V, Deodato M, Robotti M, Koutsokera M, Pozzilli V, Galati C, et al. Comorbidities of primary headache disorders: a literature review with meta-analysis. J Headache Pain. 2021;22(1):1–18.

Martelletti P. Migraine in Medicine. Migraine in Medicine. 2022.